# Intranuclear Delivery of Nuclear Factor-Kappa B p65 in a Rat Model of Tooth Replantation

**DOI:** 10.3390/ijms22041987

**Published:** 2021-02-17

**Authors:** Chung-Min Kang, Seunghan Mo, Mijeong Jeon, Ui-Won Jung, Yooseok Shin, Jin-Su Shin, Bo-Young Shin, Sang-Kyou Lee, Hyung-Jun Choi, Je Seon Song

**Affiliations:** 1Department of Pediatric Dentistry, College of Dentistry, Yonsei University, Seoul 03722, Korea; kangcm@yuhs.ac (C.-M.K.); mosh726@gmail.com (S.M.); mjjeon1107@yuhs.ac (M.J.); 2Oral Science Research Center, College of Dentistry, Yonsei University, Seoul 03722, Korea; DENSYS@yuhs.ac; 3Department of Periodontology, College of Dentistry, Yonsei University, Seoul 03722, Korea; drjew@yuhs.ac; 4Department of Conservative Dentistry, College of Dentistry, Yonsei University, Seoul 03722, Korea; 5Department of Biotechnology, College of Life Science and Biotechnology, Yonsei University, Seoul 03722, Korea; jinsuand@naver.com (J.-S.S.); pcjsby@hanmail.net (B.-Y.S.); sjrlee@yonsei.ac.kr (S.-K.L.)

**Keywords:** nuclear factor kappa B, intra-nucleus delivery, osteoclast, tooth replantation

## Abstract

After avulsion and replantation, teeth are at risk of bone and root resorption. The present study aimed to demonstrate that the intra-nuclear transducible form of transcription modulation domain of p65 (nt-p65-TMD) can suppress osteoclast differentiation in vitro, and reduce bone resorption in a rat model of tooth replantation. Cell viability and nitric oxide release were evaluated in RAW264.7 cells using CCK-8 assay and Griess reaction kit. Osteoclast differentiation was evaluated using quantitative reverse transcriptase-polymerase chain reaction (RT-PCR) and tartrate-resistant acid phosphatase (TRAP) staining. Thirty-two maxillary rat molars were extracted and stored in saline (*n* = 10) or 10 µM nt-p65-TMD solution (*n* = 22) before replantation. After 4 weeks, specimens were scored according to the inflammatory pattern using micro-computed tomography (CT) imaging and histological analyses. nt-p65-TMD treatment resulted in significant reduction of nitric oxide release and osteoclast differentiation as studied using PCR and TRAP staining. Further, micro-CT analysis revealed a significant decrease in bone resorption in the nt-p65-TMD treatment group (*p* < 0.05). Histological analysis of nt-p65-TMD treatment group showed that not only bone and root resorption, but also inflammation of the periodontal ligament and epithelial insertion was significantly reduced. These findings suggest that nt-p65-TMD has the unique capabilities of regulating bone remodeling after tooth replantation.

## 1. Introduction

Tooth avulsion is defined as the complete dislocation of tooth from the alveolar socket as a result of trauma, and it is characterized by compromised neurovascular supply, pulp necrosis, and loss of periodontal ligament (PDL) cells [1]. Tooth replantation is an acceptable option post avulsion to preserve the structures adherent to the root surface. The greatest concerns in tooth replantation are preservation of PDL vitality, cementum integrity, and minimal bacterial contamination [2,3]. The outcome of periodontal healing depends on several factors, including extra-alveolar period, storage medium, damage of PDL cells, root development, and pulp condition [4].

In order to minimize the adverse effects of avulsion, root surface treatments have been investigated for their ability to preserve PDL cell vitality [1,5]. In cases of delayed tooth replantation, necrotic remnants of PDL cells stimulate the development of external root resorption, which may lead to early tooth loss [4,6]. Several substances, such as sodium fluoride, stannous fluoride, tetracycline, citric acid, calcium hydroxide, and bisphosphonates, have been used to treat the root surface of avulsed teeth to increase their retention rate [7]. Root resorption is enhanced by substances released from inflammatory cells in the surrounding tissues, such as osteoclast activating factor, macrophage chemotactic factor, and prostaglandins [8]. Considering the root resorption characteristics, the regulation of transcription factors related to inflammatory response is an effective strategy to control inflammation-related bone and root resorption after replantation.

Nuclear factor kappa B (NF-κB) is a key mediator for both physiological immunity and pathological inflammation. NF-κB controls gene expression of inflammatory mediators, such as cytokines, chemokines, and major histocompatibility complex proteins, and has immune regulation-associated function [9]. Interestingly, inhibition of NF-κB p65 (RelA) subunit suppresses inflammation and modulates immune function; specific blockade of p65 was found to reduce osteoclastogenesis and increase bone formation [10]. Previously, nucleus-transducible form of p65-transcription modulation domain (TMD), named nt-p65-TMD, conjugated with an Hph1-protein transduction domain (PTD) was developed that could translocate efficiently into the nucleus and interfere with p65 transcription by targeting the promoters of p65-target genes [11,12,13].

This study aimed to investigate the effect of interactomic inhibition of endogenous p65 on the secretion of inflammatory cytokines and osteoclast differentiation after tooth replantation. In the present study, we examined the role of nt-p65-TMD in cellular cytotoxicity and osteoclast differentiation in vitro. In addition, we investigated the therapeutic effects of nt-p65-TMD on inflammation-related bone and root resorption after tooth replantation in a rat model.

## 2. Results

### 2.1. Generation and Transduction Efficiency of nt-p65-TMD

The N-terminal of p65 has TMD comprising DNA-binding amino acid residues and isotype-specific sequences that may play key roles in the functional specificity of p65. To modulate p65-mediated NF-κB functions, we generated transducible fusion protein (p65-TMD) of p65 with Hph-1-PTD (YARVRRRGPRR). nt-p65 was designed as a novel therapeutic to deliver p65-TMD efficiently into the nucleus of the cells in vitro and in vivo. Thus, the delivered nt-p65-TMD competitively interferes with the transcriptional activity of endogenous p65 at the promoter of p65-target genes (Figure 1A). To examine the intracellular transduction efficiency, we treated RAW264.7 cells with nt-p65-TMD. nt-p65-TMD was effectively transduced into the cells in a dose-dependent manner (Figure 1B) after 2 h treatment.

### 2.2. Viability and NO Release of nt-p65-TMD-Treated RAW264.7 Cells

Cell viability was evaluated using LY294002 treatment group as positive control. Cell viability differed significantly at 2 and 5 µM of the nt-p65-TMD treatment group (Figure 1C). NO release differed significantly at 0.5, 1, and 2 between the nt-p65-TMD treatment and LPS treatment groups (100% of LPS treatment group, Figure 1D).

### 2.3. Osteoclast Differentiation of nt-p65-TMD-Treated RAW264.7 Cells

Tartrate-resistant acid phosphatase (TRAP)/alkaline phosphatase (ALP) staining of receptor activator of nuclear factor-kB ligand (RANKL)-treated RAW264.7 cells resulted in red-stained osteoclasts. nt-p65-TMD treatment group showed lighter TRAP staining than the RANKL treatment group; moreover, the higher the nt-p65-TMD concentration, lower was the TRAP staining intensity observed (Figure 2A). The expression of several osteoclast differentiation related genes was confirmed using qPCR. TRAP, tumor necrosis factor receptor (TNFR)-associated factors (TRAF)6, Cathepsin K (CTSK), and nuclear factor of activated T cell, cytoplasmic 1 (NFATc1) expression was lower in nt-p65-TMD treatment group, and differed significantly compared to that in the RANKL treatment group (100% of RANKL treatment group, Figure 2B).

### 2.4. Replantation of Rat Maxillary First Molar

Radiographic analysis revealed an extensive (score 3) inflammatory absorption of alveolar bone (mean score = 2.4) in saline group. Root absorption was further progressed to entire absorption of cementum and dentin (score 3) in many cases (mean score = 2.5). In the nt-p65-TMD group, bone resorption showed a relatively small level of resorption with overall absorption (average = 1.68). Root resorption was mild or limited to the surface in most cases (average = 1.86). Although cellular cementum destruction occurred in most cases of both the groups, differences in the degree of inflammatory alveolar bone and root resorption between the groups were observed. While some teeth did not show root resorption, localized inflammatory root resorption and alveolar bone resorption occurred in most cases (Figure 3C,F). In the saline group, root and bone resorption occurred more extensively, and resorption to dentin beyond the surface absorption of cementum was observed in several cases (Figure 3B,E). Bone resorption between the groups differed significantly (*p* < 0.05), whereas the difference in root resorption was not statistically significant (Figure 3G,H). Further, the difference in replacement resorption was not statistically significant between the saline (mean = 0.30) and nt-p65-TMD (mean = 0.18) groups (Figure 3I). Similarly, no statistical difference was observed in the degree of pulp mineralization between the saline (mean = 0.40) and nt-p65-TMD (mean = 0.38) groups (Figure 3J).

Histological analysis revealed a capsular layer of inflammatory cells and fibrous tissues that suppressed the adhesion of PDLs in the saline group. In some cases, resorption was severely progressed, and PDLs and fibroblasts were rarely observed; inflammatory cells, including a large number of lymphocytes, plasma cells, and neutral multinuclear cells, were observed; whereas the apical third was completely lost (Figure 4B,E). In the nt-p65-TMD group, the PDL space was expanded due to partial alveolar bone resorption, but adhesion of the fibers was relatively good, and the arrangement of PDLs and fibroblasts was partially recovered (Figure 4C,F). Although the arrangement of PDL fibers was irregular, the fibers were attached to the newly formed cementum and showed healing. Moreover, differences in the degree of alveolar bone resorption, root resorption, inflammation in the PDL, and inflammation at the epithelial insertion were statistically significant (*p* < 0.01) between the two groups (Figure 4G–J).

## 3. Discussion

In this study, we confirmed a novel therapeutic strategy to suppress inflammation and osteoclastogenesis by intranuclear delivery of the TMD of p65 in vitro and in vivo. nt-p65-TMD, a fusion protein between TMD of p65 and a human origin Hph-1-PTD, could be delivered effectively into the nucleus without influencing the cell viability. nt-p65-TMD significantly reduced osteoclast differentiation and inflammatory response after replantation.

nt-p65-TMD was previously reported to be effective in reducing inflammation. nt-p65-TMD-mediated NF-κB inhibition was reported to have an inhibitory effect on severe sepsis and inflammasome after surgery [11,13]. NF-κB is a transcription factor that plays an important role in the inflammatory response, and nt-p65-TMD affects the inflammatory response via inhibition of NF-κB activity. Exposure to bacterial products, such as LPS, enhances NO production in RAW264.7 cells [14,15]. In the present study, the production of NO in the nt-p65-TMD group decreased in a dose-dependent manner, which was more effective at lower concentrations of nt-p65-TMD (0.5 and 1 µM). However, the inflammatory response did not decrease below 50% in the NO assay; this may be attributed to the fact that inflammation is induced and regulated through several pathways.

Further, nt-p65-TMD reduced osteoclast differentiation by inhibiting NF-κB. RANKL, expressed by osteocytes, is required for the differentiation of osteoclast precursors into mature osteoclasts [16,17,18]. The released RANKL binds to its receptor RANK, a member of the TNFR superfamily lacking the intrinsic enzymatic activity that is required for activating downstream signaling molecules [19]. Thus, RANK transduces intracellular signals by recruiting adaptor molecules, such as TRAFs, that then activate mitogen activated protein kinases (MAPKs) and NF-κB [20,21,22]. TRAF6 is the major adaptor protein responsible for mediating RANKL-activated signaling cascades. Activated TRAF6 promotes NF-κB activity [23]. Decreased expression of TRAF6 leads to suppression of NF-κB function. TRAF6-deficient mice showed severe osteopetrosis [24,25]. Further, RANK signaling finally induces the amplification of NFATc1 and expression of osteoclastic genes [26]. Moreover, NFATc1-knockout mice exhibit osteopetrosis and inhibit osteoclastogenesis in vitro and in vivo [27]. In the nt-p65-TMD group of the present study, the expression of TRAF6 and NFATc1 decreased even after RANKL treatment. Eventually, nt-p65-TMD inhibited the function of NF-κB, resulting in inhibition of osteoclast differentiation.

In vivo study revealed that nt-p65-TMD can affect tooth replantation via two processes: Inflammation modulation and bone remodeling. First, reduced secretion of the pro-inflammatory cytokine might affect the inflammation-related resorption. After tooth avulsion, the dentinal tubules act as entry-points for bacteria and toxins, causing an inflammatory reaction in the PDL and destroy the root and alveolar bone [28]. In the saline group of our study, we observed a lack of periodontal repair and occurrence of root resorption in many cases after replantation. These histological observations reportedly accompany inflammatory changes in the PDL along with the resorption of cementum and dentin, and inflammation was detected in the granulation tissue in the PDL that includes a large number of lymphocytes, plasma cells, and neutral multinuclear cells [29,30]. Conversely, in the nt-p65-TMD group, inflammation was found as a local feature of surface resorption in the cementum. Interestingly, histological analysis of the nt-p65-TMD group revealed the occurrence of partial recovery of PDLs and fibroblasts. This finding was similar to the previous study in which dexamethasone was applied locally to the replanted teeth of rats [31]. The study demonstrated that local dexamethasone application has anti-inflammatory capacity to control the initial inflammatory response and allows recovery by cementoblast.

In vitro study confirmed that nt-p65-TMD treatment (0.5 µM) caused significant reduction in NO release (30%) by inhibiting inflammation, thus indicating the involvement of subunit p65 in inflammation around the replanted teeth. Previous studies have reported the relationship between NF-κB and the protein that forms inflammasome, which is related to the inflammatory response activity [32]. In human monocytes, NLRP3 induces NF-κB activity via TLR pathway in bacterial inflammation [33]. In addition, nt-p65-TMD used in the current study was reported to inhibit the formation of inflammasome complex as well as expression of related cytokines after surgery [13]. Therefore, nt-p65-TMD was expected to effectively suppress inflammation during tooth replantation in this study.

Next, in vivo study confirmed that nt-p65-TMD affects osteoclast differentiation by inhibiting the function of NF-κB. Reducing the osteoclastic activity is another potential approach for post traumatic treatment. Cell damage resulting from trauma stimulates an inflammatory response that results in the osteoclast-mediated external root resorption [34]. Similar to our findings, inhibition of bone resorption was observed after topical application of alendronate to the replanted teeth of rats [35,36], suggesting an inhibition of osteoclast function to be the main function of alendronate. As nt-p65-TMD inhibits not only bone resorption, but also root resorption, it may also influence the function of odontoclasts. Further studies are needed to investigate how nt-p65 affects the cells around the replanted teeth.

The present study has several limitations. First, we did not confirm whether nt-p65-TMD was well transduced into the tooth surface and alveolar socket after tooth replantation. However, in previous experiments using Hph-1-PTD domain, the proteins were reportedly well transduced to various cells and tissues, particularly through the skin [37]. Therefore, nt-p65-TMD might have sufficiently transduced into the tooth and bone structure in the present study. Of course, it may be questionable whether application to the tooth surface for only 5 min can result in long-term effects. However, saline or nt-p65-TMD was applied to the alveolar socket as well as to the extracted tooth surface, which could be maintained even after replantation. It will be more meaningful to add an experiment for studying the effects over time post-application. The second, long-term effect of nt-p65-TMD treatment on ankylosis and canal calcification was not evaluated. Further follow-up studies should be performed to confirm the practical clinical applications of this study. Third, in order to effectively process nt-p65-TMD, its optimal concentration of the treatment dose needs to be determined through further experimentation. Moreover, it needs to compare favorably with other anti-inflammation-related transcription factor.

## 4. Materials and Methods

### 4.1. Production of Nucleus-Transducible Form of p65-TMD

To generate DNA construct of nt-p65-TMD, DNA sequences encoding the N-terminus of p65 (a.a 1-187) and Hph-1-PTD were amplified by PCR and inserted into the pET-28a(+) vector (Novagen Merck Millipore, Billerica, MA, USA). The DNA construct was transformed into Escherichia coli BL21 CodonPlus (DE3)-RIPL strain (Invitrogen, Carlsbad, CA, USA), and protein expression was induced for 5 h at 37 °C with 1 mM isopropyl-β-D-thiogalactopyranoside (GenDEPOT, Barker, TX, USA). After sonicating the harvested cells in lysis buffer (10 mM imidazole, 50 mM NaH2PO4, 300 mM NaCl, pH 8.0), a soluble fraction of the lysates was separated by centrifugation (12,000 rpm for 10 min at 4 °C). The soluble proteins were loaded into a HisTrap HP chromatography column (GE Healthcare Life Sciences, Marlborough, MA, USA) using Acta pure (GE Healthcare Life Sciences, Marlborough, MA, USA), and protein-bound chromatography column was washed with wash buffer (40 mM imidazole, 50 mM NaH2PO4, 300 mM NaCl, pH 8.0). Proteins were eluted with elution buffer (500 mM imidazole, 50 mM NaH2PO4, 300 mM NaCl, pH 8.0). The eluted proteins were loaded into a HiTrap SP HP chromatography column in ion-exchange chromatography binding buffer (50 mM NaH2PO4, pH 6.0) using Acta pure. Bound proteins were washed with SP wash buffer (50 mM NaH2PO4, 300 mM NaCl, pH 6.0), and then target proteins were eluted with SP elution buffer (50 mM NaH2PO4, 1 M NaCl, pH 7.0). The buffer containing eluted proteins was exchanged into 10% glycerol PBS using PD-10 Sephadex G25 (GE Healthcare Life Sciences, Marlborough, MA, USA).

### 4.2. In Vitro Study

#### 4.2.1. Cell Culture and Cell Viability

RAW264.7 cells were obtained from the Korean Cell Line Bank and cultured in Dulbecco’s Modified Eagle Medium (DMEM; Invitrogen, Carlsbad, CA, USA) containing 10% fetal bovine serum (FBS; Invitrogen, Carlsbad, CA, USA), 100 U/mL penicillin, 100 µg/mL streptomycin (Invitrogen, Carlsbad, CA, USA), and 0.2% amphotericin B (Invitrogen) at 37 °C in 5% CO_2_. To evaluate the cell viability, RAW264.7 cells were plated at a concentration of 25,000 cells/well in a 24-well plate. After 24 h, cells were treated with nt-p65-TMD or LY294002 for 2 h. After 24 h, the quantity of water-soluble colored formazan from the Cell Counting Kit (CCK)-8 assay (Dojindo Laboratories, Kumamoto, Japan) formed by the activity of dehydrogenases in living cells was measured using a spectrophotometer (Benchmark Plus microplate spectrophotometer, Bio-Rad Laboratories Inc., Hercules, CA, USA) at 450 nm. In this study, LY294002 was used as a positive inhibitor of NF-kB. There are various inhibitors that inhibit the activity of NF-kB, one of which is LY294002. At this time, LY294002 acts as a PI3K inhibitor. PI3K is an upper transcription factor of NF-kB, and it has been shown that inhibition of PI3K activity inhibits the activity of NF-kB [38,39]. Cell viability assay data were obtained from three independent experiments, with all samples run in triplicate.

#### 4.2.2. Western Blot

After nt-p65-TMD treatment in a dose-dependent manner for 2 h, RAW264.7 cells were lysed and the cell lysate was electrophoresed and transferred onto polyvinylidene difluoride (PVDF) membranes (Bio-Rad Laboratories Inc., Hercules, CA, USA). The membrane was blocked using blocking buffer (4% bovine serum albumin in Tris-buffered saline containing Tween 20 (TBST) to prevent non-specific binding of antibodies. Blocked membrane was incubated with anti-DYKDDDDK (FLAG) or anti-β-actin antibody (Cell signaling, Beverly, MA, USA) at 4 °C overnight. After TBST wash, anti-mouse IgG or anti-rabbit IgG (Abcam, Cambridge, UK) was used to detect each primary antibody. For the chemiluminescence reaction, ECL reagent (Bio-Rad Laboratories Inc., Hercules, CA, USA) was added onto the membrane and the signal was observed using ChemiDoc (Bio-Rad Laboratories Inc., Hercules, CA, USA.).

#### 4.2.3. Nitric Oxide (NO) Assay

RAW264.7 cells were seeded into 24-well plates at a density of 2.5×104 cells/well. After nt-p65-TMD or LY294002 treatment in dose-dependent manner for 2 h, the cells were incubated with LPS (0.1 µg/mL) for 20 h at 37 °C in CO_2_ incubator. To determine the nitrite release in the culture media, presumed to reflect the NO levels, Griess reaction was used, wherein 100 µL cell culture medium was mixed with 100 µL Griess reagent (Invitrogen) and incubated at room temperature for 30 min. The NO concentration was determined at 540 nm using a spectrophotometer (Bio-Rad Laboratories Inc., Hercules, CA, USA)

#### 4.2.4. Osteoclast Differentiation

RAW264.7 cells were seeded onto 24-well plates at a density of 1×10^4^ cells/well and treated with nt-p65-TMD or LY294002 for 2 h; DMEM was replaced with Minimum Essential Medium (MEM) alpha (Invitrogen, Carlsbad, CA, USA) containing 10% FBS and 10 ng/mL of receptor activator of nuclear factor-kB ligand (RANKL) (PeproTech Inc., Rocky Hill, NJ, USA), and the cells were incubated for 3 days at 37 °C in CO_2_ incubator. Tartrate-resistant acid phosphatase (TRAP) staining was performed using TRAP/ALP staining kit (WAKO, Osaka, Japan). Total RNA was isolated using RNeasy Mini Kit (Qiagen, Valencia, CA, USA) according to manufacturer’s instructions. Integrity and concentration of the extracted RNA was evaluated using a spectrophotometer (NanoDrop ND-2000, Thermo Scientific, Waltham, MA, USA). Next, 500 ng RNA aliquots were reverse transcribed to cDNA using a Maxime RT premix kit [oligo d(T)15 primer; Intron Biotechnology, Seongnam, Gyeonggi, Korea] according to manufacturer’s instructions. A quantitative real-time PCR (qPCR) assay was performed with SYBR Premix Ex Taq (Takara Bio Inc., Otsu, Japan) and a real-time PCR system (ABI 7300, Applied Biosystems, Carlsbad, CA, USA) as per manufacturer’s instructions. The qPCR conditions were 95 °C for 10 sec followed by 40 cycles of 95 °C for 5 sec and 60 °C for 30 sec, with a final 5-min extension at 72 °C. Expression for each gene was normalized to that of glyceraldehyde-3-phosphate dehydrogenase (GAPDH), and the relative expression levels of the target genes were calculated using the 2–ΔΔCt method (Livak and Schmittgen 2001). The specific primers for each gene are listed in Table 1.

### 4.3. In Vivo Study

#### 4.3.1. Replantation of Rat Maxillary First Molar

The procedure for replantation of rat maxillary first molar was performed in accordance with protocol approved by the Institutional Animal Care and Use Committee of Yonsei University (#2018-0214). nt-p65-TMD, diluted to 10 µM in saline, was used for the in vivo study. Thirty-two 6-week male Sprague Dawley rats (Orient Bio, Seoul, Korea) weighing 200–250 g were used in this study. To facilitate the extraction, 0.4% β-aminoproprionitrile (Sigma, St. Lois, MO, USA) mixed in distilled water was administered to the rats along with the feed 3 days prior to extraction. The maxillary left right molar was atraumatically extracted under anesthesia with ketamine (0.1 mL/100 g, Yuhan, Dongjak, Seoul, Korea) and xylazine (0.05 mL/100 g, Bayer Korea, Dongjak, Seoul, Korea). After performing peritomy using Explorer, the tissue was extracted with minimal trauma from the alveolar socket using tissue forceps. To determine the drug effect on PDL healing, the control and experimental groups were divided, based on the teeth treatment, as follows: In the control group (*n* = 10), the extracted teeth were stored in 50 mL of 0.9% physiological saline (pH 6.0) at 4 °C for 5 min, and cotton soaked with 0.9% physiological saline was applied to the alveolar socket to induce thrombus removal and hemostasis. In the experimental group (*n* = 22), the extracted teeth were stored in 10 µM p65-TMD solution (pH 7.22) at 4 °C for 5 min, and cotton soaked with 10 µM nt-p65-TMD solution was applied to the alveolar socket. Immediately after the root surface treatment, the teeth were replanted in the original sockets. After replantation, all animals received a single intramuscular dose of 20,000 UI penicillin G benzathine (Tokyo Chemical Industry Co. Ltd., Tokyo, Japan). Animals in both groups were sacrificed under ketamine anesthesia 4 weeks after replantation. The maxilla was removed, washed with saline, and fixed in 10% formalin.

#### 4.3.2. Micro-Microcomputed Tomography (CT) Image Analysis

The extracted maxilla was scanned at 10 µm intervals using a micro-CT system (Quantum FX micro-CT, Parkin Elmer, Norwalk, CT, USA), and the tooth images were reconstructed using a software (TRI-3D, Ratoc System Engineering Co., Ltd., Tokyo, Japan). As mesiobuccal (MB) root is the largest and allows clear observation of changes, it was selected. A longitudinal image of the replanted tooth was obtained through long axis of the MB root in the mesiodistal direction. In the longitudinal image, the following events were investigated: A. Bone resorption (The extension of bone surface resorption area in contact with the root was measured.); B. Root resorption; C. Ankylosis (direct fusion of bone to root); D. Pulp mineralization. To quantify the changes in bone and root resorption, scores were attributed to the different events listed below (Table 2). The criteria defined in the study for scoring in micro-CT image analysis and histological examination were in accordance with those described by Poi et al. [40].

#### 4.3.3. Histological Examination

After the micro-CT scan, the maxilla was decalcified with 10% ethylenediaminetetraacetic acid (pH 7.4; Fisher Scientific Co., Houston, TX, USA) for 3 weeks at room temperature. The degree of decalcification was confirmed by radiographical analysis. After cryotomy, 20–25-μm thick section of maxillary first molar was cut from the sagittal plane to the apical axis, and tissues well connected with the apical and apical roots were selected, stained with hematoxylin and eosin, and observed using an optical microscope (Axio Lab, Zeiss, Germany). The degree of inflammatory resorption of the roots, degree of substitutional substitution, PDL status, and degree of inflammation were observed for each group using an optical microscope. For quantitative analysis, in the same as the micro-CT image analysis, only MB root was observed. In the longitudinal section, the following events were investigated: A. Bone resorption; B. Root resorption; C. Inflammation at the epithelial insertion; D. Inflammation in the PDL (Table 3).

### 4.4. Statistical Analyses

Statistical analysis was performed with SPSS (version 25.0, SPSS Inc., Chicago, IL, USA). All in vitro experiments were performed at least in triplicate. The normality of the in vitro data was evaluated using the Shapiro–Wilk test (*p* > 0.05). Multiple comparison testing was conducted using the ANOVA (*p* < 0.05), followed by the t-test for comparing control and experimental group (*p* < 0.05). The normality of in vivo study data was evaluated using the Shapiro–Wilk test (*p* < 0.05). Mann–Whitney U test (*p* < 0.05) for comparing saline and nt-p65-TMD group was performed with SPSS.

## 5. Conclusions

In conclusion, nt-p65-TMD can be a novel therapeutic to prevent root and bone resorption after tooth replantation. The present study is the first to demonstrate the possibility of intranuclear delivery of synthetic nuclear factor to the tissue in the healing process after tooth replantation. Furthermore, the strategy can be applied for the development of novel therapeutics for bone remodeling issues in dentistry, especially where a specific transcription factor is known to have a key role in the pathogenesis.

## Figures and Tables

**Figure 1 ijms-22-01987-f001:**
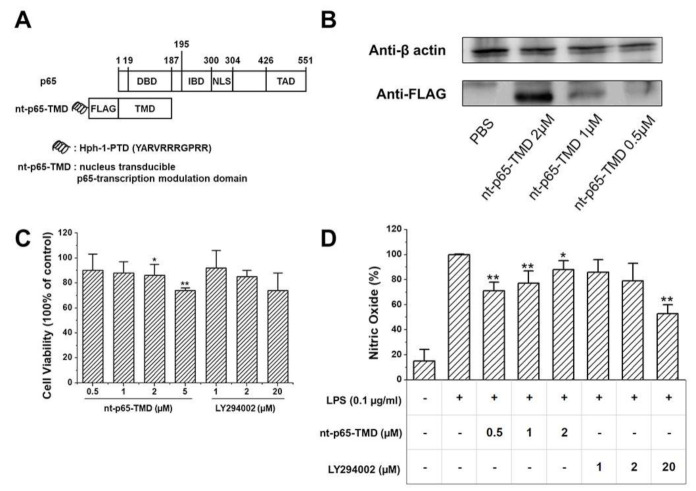
(**A**) Structure of nt-p65-TMD. (**B**) Transduction efficiency of nt-p65-TMD. Dose-dependent intracellular transduction efficiency of nt-p65-TMD into RAW264.7 cells. nt-p65-TMD was detected by immunoblot analysis with anti-FLAG antibody. β-actin was used as loading control. (**C**) Cell viability of nt-p65-TMD in RAW264.7 cells. Cells were treated with different concentrations of nt-p65-TMD for 24 h, and cell viability was assessed using CCK-8 assay kit. LY294002 treatment groups were considered positive controls. The data are expressed as mean ± standard deviation. Cell viability differed significantly for the 2 and 5 µM nt-p65-TMD treatment groups compared to the control group. Data show mean ± standard deviation values of three independent experiments. * *p* < 0.05 and ** *p* < 0.01 indicate significant differences compared to the control value (100%). (**D**) NO release of nt-p65-TMD in RAW264.7 cells. Cells were incubated in the presence of different concentrations of nt-p65-TMD and 0.1 µg/mL LPS for 20 h. Then, the culture supernatant was analyzed for NO. Data show mean ± standard deviation values of three independent experiments. * *p* < 0.05 and ** *p* < 0.01 indicate significant differences compared to LPS-stimulation value (100%).

**Figure 2 ijms-22-01987-f002:**
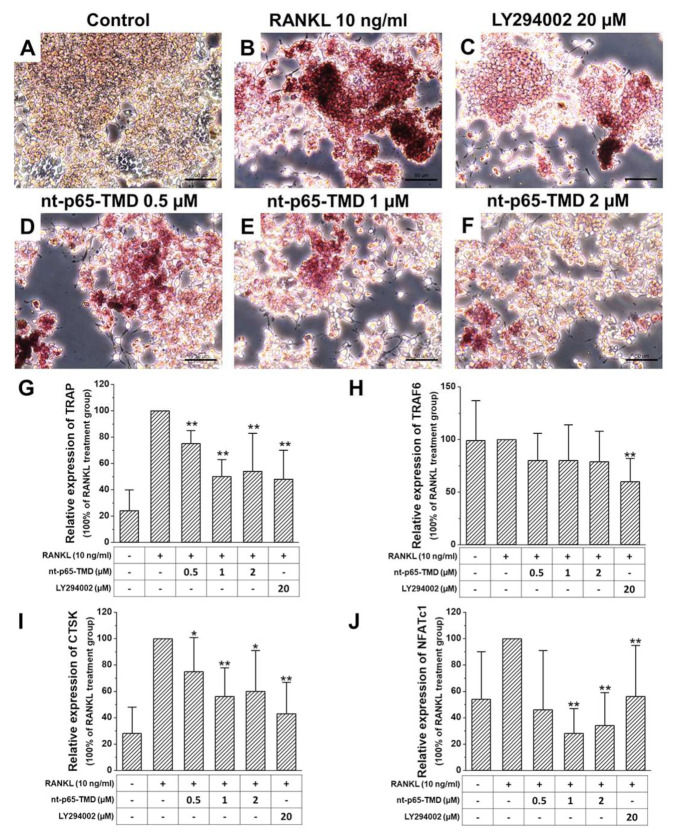
Osteoclast differentiation of nt-p65-TMD in RAW264.7 cells (**A**–**F**) tartrate-resistant acid phosphatase (TRAP) staining of nt-p65-TMD in RAW264.7 cells. Cells were pretreated with different concentrations of nt-p65-TMD for 2 h followed by treatment with 10 ng/mL receptor activator of nuclear factor-kB ligand (RANKL) for 3 days. Subsequently, RAW264.7 cells were fixed and stained to detect TRAP. The osteoclasts were stained red. Scale bars = 50 µm (**G**–**J**) Changes in the expression of the osteoclast-related genes of nt-p65-TMD in RAW264.7 cells. Cells were pretreated with different concentrations of nt-p65-TMD for 2 h followed by treatment with 10 ng/mL RANKL for 3 days. RNA was isolated from RAW264.7 cells and cDNA was synthesized. Expression of tartrate-resistant acid phosphatase (TRAP), TNF receptor associated factor 6 (TRAF6), CathepsinK (CTCK), and nuclear factor of activated T cell, cytoplasmic 1 (NFATc1) was evaluated using quantitative RT-PCR relative to RANKL treatment group (normalized to 100%). Data were obtained from five separate experiments, with all samples run in duplicate. The data are expressed as mean ± standard deviation values. The expression of the genes differed significantly; t-test and Mann–Whitney U test, * *p* < 0.05, ** *p* < 0.01.

**Figure 3 ijms-22-01987-f003:**
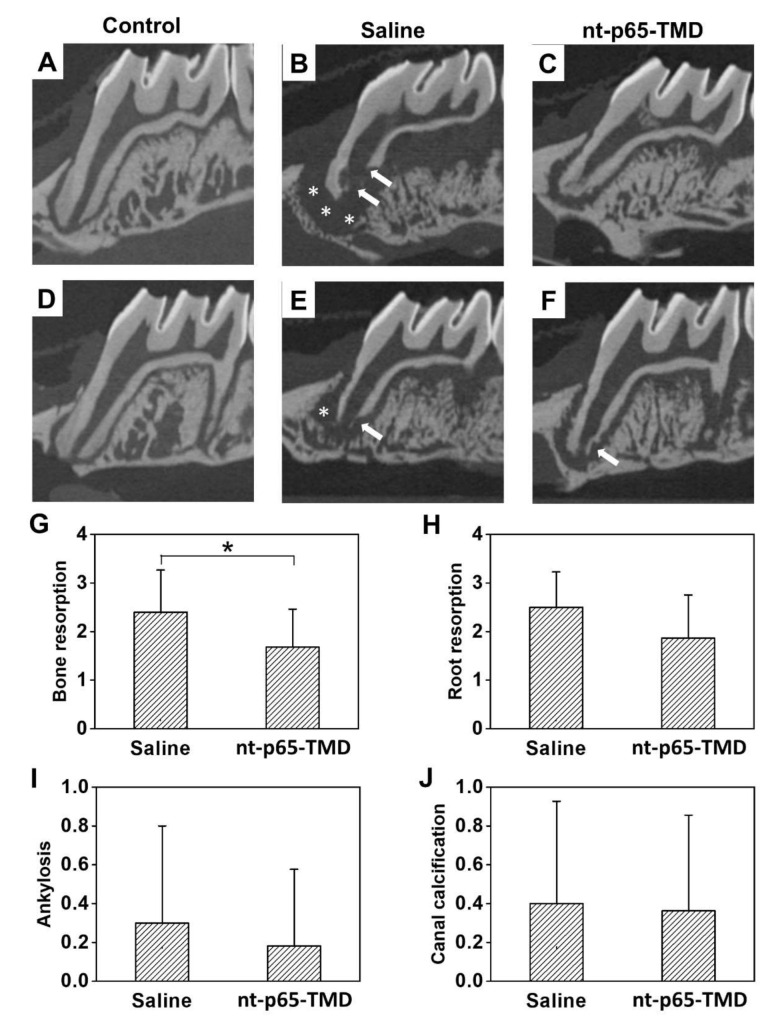
Two-dimensional horizontal microcomputed tomography images of rat maxillary first molars 4 weeks after replantation. (**A**,**D**) Control: The teeth without extraction and replantation. Both bone and root resorption was not observed. (**B**,**E**) Saline group: 0.9% physiological saline was applied to the extracted teeth surface and alveolar socket. Severe, extensive bone (white asterisk) and root resorption (arrow) were observed. (**C**,**F**) nt-p65-TMD group: 10 µM p65-TMD solution was applied to the surface of the extracted teeth and alveolar socket. A relatively short range of bone resorption (white asterisk) was observed compared to that in the saline group. (**G**–**J**) Evaluated scores, represented as mean ± standard deviation, for (**G**) bone resorption, (**H**) root resorption, (**I**) ankylosis, and (**J**) pulp mineralization in the saline and nt-p65-TMD groups. Based on the scores of the four categories, only bone resorption showed a significant decrease in the nt-p65-TMD group; Mann–Whitney U test, * *p* < 0.05.

**Figure 4 ijms-22-01987-f004:**
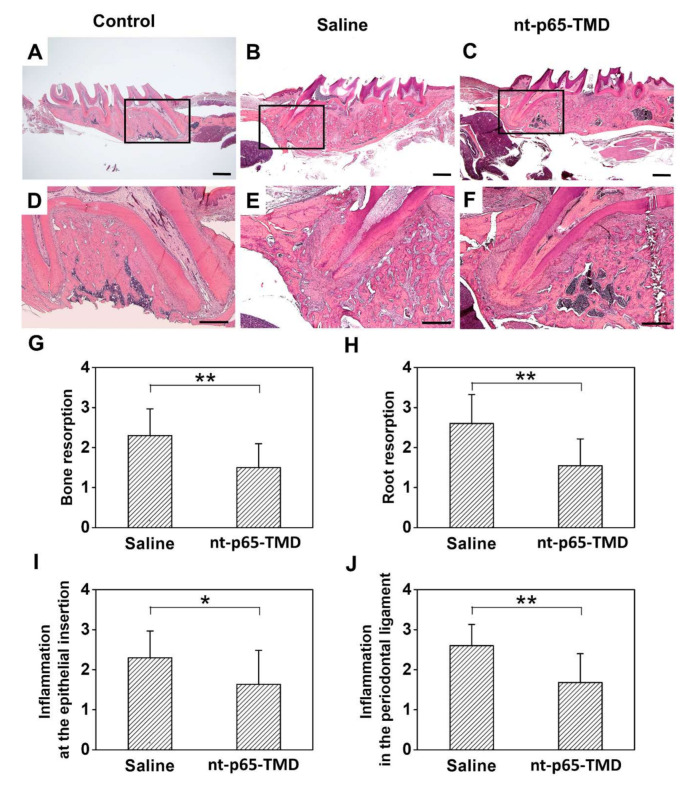
Histological analysis of replanted rat maxillary first molars 4 weeks after replantation. (**A**,**D**) Control group: The teeth without extraction and replantation. Inflammatory root resorption and bone resorption was not observed. (**B**,**E**) Saline group: The extracted teeth were replanted after storing in 0.9% physiological saline for 5 min. A large number of inflammatory cells were observed, and the apical third was lost by severe inflammatory root resorption. (**C**,**F**) nt-p65-TMD group: Localized root resorption and partial recovery of periodontal ligament and fibroblasts were observed (**G**–**J**) Evaluated scored, expressed as mean ± standard deviation, for (**G**) bone resorption, (**H**) root resorption, (**I**) inflammation at the epithelial insertion, and (**J**) inflammation in the periodontal ligament in the saline and nt-p65-TMD groups. Hematoxylin-eosin staining, Scale bars = 1000 µm (applies to **A**–**C**) and 500 µm (applies to **D**–**F**); Mann–Whitney U test, * *p* < 0.05, ** *p* < 0.01 (applies to **G**–**J**).

**Table 1 ijms-22-01987-t001:** qPCR forward and reverse primer sequences. The annealing procedures were performed at 60 °C for all primers.

Gene	Forward Primer Sequence (5′–3′)	Reverse Primer Sequence (5′–3′)
*CTSK*	GGGATGTTGGCGATGCA	CCAGCTACTTGAGGTCCATCTTC
*NFATc1*	CACTGGCGCTGCAACAAGA	CATTCCGGAGCTCAGCAGAATAA
*TRAF6*	ACCTGAACGCGCCTTCTG	CATCCAGCTGACTCGTTTCATAA
*TRAP*	CAAAGGTGCAGCCTTTGTGTC	TCACAGTCCGGATTGAGCTCA
*GAPDH*	CTGGCACAGGGTATACAGGGTTAG	ACTGGTGCCGTTTATGCCTTG

Abbreviations: CTSK, gene encoding cathepsin K; NFATc1, gene encoding nuclear factor of activated T-cells, cytoplasmic 1; TRAF6 gene encoding TNF receptor associated factor 6; TRAP; gene encoding tartrate-resistant acid phosphatase, GAPDH, gene encoding glyceraldehyde-3-phosphate dehydrogenase.

**Table 2 ijms-22-01987-t002:** Scores for quantitative analysis in micro-microcomputed tomography image.

Event	Score	Characteristics
Bone resorption	0	No resorption present
	1	Resorption occurred in <1/3 of the bone surface
	2	Resorption occurred in >1/3 and <2/3 of the bone surface
	3	Resorption occurred in >2/3 of the bone surface
Root resorption	0	No resorption present
	1	Resorption occurred in <1/3 of the root surface
	2	Resorption occurred in >1/3 and <2/3 of the root surface
	3	Resorption occurred in >2/3 of the root surface
Ankylosis	0	No ankylosis present
	1	Ankylosis present
Pulp mineralization	0	No pulp mineralization present
	1	Pulp mineralization present

**Table 3 ijms-22-01987-t003:** Scores for quantitative analysis in histological examination.

Event	Score	Characteristics
Bone resorption	0	No resorption present
	1	Resorption occurred in <1/3 of the bone surface
	2	Resorption occurred in >1/3 and <2/3 of the bone surface
	3	Resorption occurred in >2/3 of the bone surface
Root resorption	0	No resorption present
	1	Resorption occurred in <1/3 of the root surface
	2	Resorption occurred in >1/3 and <2/3 of the root surface
	3	Resorption occurred in >2/3 of the root surface
Inflammation at the epithelial insertion	0	Absence or occasional presence of inflammatory cells
	1	Inflammatory process restricted to lamina propria of the internal part of epithelium
	2	Inflammatory process extending apically to the small portion of the connective tissue underlying the lamina propria of the internal portion of the gingival epithelium
	3	Inflammatory process reaching the proximity of the alveolar bone crest
Inflammation in the PDL	0	Absence or occasional presence of inflammatory cells
	1	Inflammatory process present only in the apical, coronal or small lateral area of PDL
	2	Inflammatory process reaching more than half of the lateral PDL of the root
	3	Inflammatory process in the whole PDL

## Data Availability

Data is contained within the article.

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
