# Peer review of "Intranuclear Delivery of Nuclear Factor-Kappa B p65 in a Rat Model of Tooth Replantation"

_ijms, 2021, doi:10.3390/ijms22041987_

Round 1

Reviewer 1 Report

The authors wrote „Cells were incubated in the presence of different concentration of nt- p65-TMD and 0.1 μg/mL LPS for 20 h “

The LPS induced activation of p65 was reported mostly within 45 min and recovered to basal level within 24 h. is the 20h treatment reasonable? 

In Figure 2 (G-J) Authors concluded that the NO production is reduced in dependence on nt-p65-TMD concentration. However only two concentrations are supporting this (0.5 and 1) which might be too weak to make this conclusion.  The data of RANKL were shown without error bars, is it a single measurement?

Figure 3, (A-F) could the authors indicate more in details about the differences in the images e.g. using arrows or circles (such as in Fig. 4).

Figure 3 (H-J) the standard deviations and error bars indicated very large deviation among the tests. Please describe/ explain/ discuss these results more in details. What is the correlation between the conclusion and these “no significant data”, or these data will not affect the main findings of this article?  If the conclusion will not be affected, are these experiments the proper experiments supporting the idea and conclusion? The authors claim that the nt-p65-TMD should inhibit both bone resorption and root resorption which cannot be supported by the data presented in figure 3.

Author Response

I would like to express my appreciation for your advice.

We agreed with your comments and introduced corrections where required in the manuscript.

I look forward to your response and hope that the revised manuscript is suitable for publication in your journal.

Yours sincerely, Je Seon Song

Reviewer 2 Report

This paper show that the effect of nt^p65-TMD for osteoclast formation and bone resorption in vitro and in vivo. The paper is important for bone biology. But, I have comments.

In vitro osteoclast experiment, the authors treated RAW cells with nt-p65-TMD for 2h. And they washed out nt-p65-TMD and checked osteoclast differentiation after 3days culture. I think RAW cells divided many times for 3 days. I think nt-p65-TMD affected only first seeded cells. I think only small number cells affected by nt-p65-TMD for osteoclast differentiation. It is difficult to believe the results.

In vivo experiment, the authors treated extracted tooth with nt-p65-TMD only for 5min. I think nt-p65-TMD affected only surface cells on tissue of extracted tooth. I think there is small effect. And the source of osteoclast is hematopoietic cell from other parts. It is difficult to believe the results.

And there are only mononuclear TRAP positive cells not multi-nuclear osteoclasts in Fig 2. The authors have to make osteoclasts. Mononuclear TRAP positive cells usually express CTSK very low. It is difficult to believe these results.

The quality of histological sections is very low. How did the authors evaluate bone and root resorption and inflammations?

Author Response

(The authors gave the same response as above.)

Round 2

Reviewer 2 Report

Point 1

They answered my comment. They answered that nt-p65-TMD is a protein that can enter and perform its functions in the nucleus. The 2-h time period used in this study was adequate for the protein to enter the nucleus, and in previous studies, when the TMD-bound protein was used to treat the cells, the presence of the protein in the nucleus was observed for up to 48 h. I think this is correct.

 My comment is that nt-p65-TMD is protein. Protein can not divide in the cells after cell division. nt-p65-TMD is not divided in the cells after cell division. RAW cells divided many times for 3 days. This means nt-p65-TMD is not divided in the cells after cell division. Therefore, nt-p65-TMD affected only first seeded cells. I think only small number cells affected by nt-p65-TMD for osteoclast differentiation. It is difficult to believe the results.

For this reason, it is difficult to experiment with osteoclasts to observe the temporary effects of proteins. If the authors want to prove, the authors use retrovirus or lentivirus vector and transfect osteoclast precursors for constantly expression of nt-p65-TMD for affection of osteoclast formation.

Point 2

In vivo experiment is same as point 1.

Point 3

They answered my comment. They answered that nt-p65-TMD may exert a significant effect on the inhibition of osteoclast differentiation even for mononuclear osteoclasts, and the results of the 3-day differentiation experiment were reported in the paper. This is not inhibitory effect of osteoclast differentiation. The authors have to add 5-day or more differentiation experiment. The authors have to add inhibitory effect of differentiation of multinuclear osteoclast.

The gene expression of osteoclast related cytokine is also same comment.

Point 4: The quality of histological sections is very low.

The authors have to change high quality figure.

Especially, point 1-2 is fundamental problem. It is difficult to recommend to publishing it.

Author Response

I would like to express our heartfelt appreciation for your delicate and realistic advice.

We have done our best to answer your questions about the intracellular expression of nt-p65-TMD.

We sincerely hope that these efforts will answer your advice.

I look forward to your response that our revised manuscript is suitable for publication in this journal.

Yours sincerely,

Je Seon Song

Round 3

Reviewer 2 Report

About point 1

They did not answer my suggestions.

It is difficult to experiment with osteoclasts to observe the temporary effects of proteins. Therefore, many researchers have a hard time to express protein constantly in osteoclast precursors during osteoclastogenesis, because there is cell proliferation. Therefore, many researchers use retrovirus or lentivirus vector and transfect osteoclast precursors for constantly expression of proteins in osteoclast formation. It is difficult to believe the results.

The authors showed the former papers, but they did not prove expression of nt-p65-TMD in osteoclast precursors in osteoclast formation.

The authors need to prove the presence of protein after osteoclast formation.This is easy experiment. The authors have to do the experiment.

About point 3

There are only mononuclear TRAP positive cells not multi-nuclear osteoclasts. The authors have to make osteoclasts. This is also easy experiment.

・The authors need to prove the presence of protein after osteoclast formation.

・The authors have to make osteoclasts.

The authors have to add two easy experiments. I think it takes for 1week.

Author Response

Once again, we sincerely appreciate your advice to thoroughly verify the results of our paper and ensure its completeness.

We received 2 months from the editor to perform additional experiments towards the revision. We apologize for the delayed reply, and we sincerely hope that our efforts can answer your questions.

Yours sincerely,

Je Seon Song
